# Exploring the Functional Heterogeneity of Directly Reprogrammed Neural Stem Cell-Derived Neurons via Single-Cell RNA Sequencing

**DOI:** 10.3390/cells12242818

**Published:** 2023-12-11

**Authors:** Yoo Sung Kim, NaRi Seo, Ji-Hye Kim, Soyeong Kang, Ji Won Park, Ki Dae Park, Hyang-Ae Lee, Misun Park

**Affiliations:** 1Advanced Bioconvergence Product Research Division, National Institute of Food and Drug Safety Evaluation, Ministry of Food and Drug Safety, Cheongju-si 28159, Republic of Korea; ysungkim@korea.kr (Y.S.K.); tj0943@korea.kr (N.S.); jhkim0304@korea.kr (J.-H.K.); sykang1215@korea.kr (S.K.); parkgeeone@korea.kr (J.W.P.); kidae@korea.kr (K.D.P.); 2Department of Predictive Toxicology, Korea Institute of Toxicology, Daejeon 34114, Republic of Korea; vanessa@kitox.re.kr

**Keywords:** direct reprogramming, neural stem cells, iNSC-derived neuron, functional heterogeneity, single-cell RNA sequencing, surrogate biomarker, quality control

## Abstract

The therapeutic potential of directly reprogrammed neural stem cells (iNSCs) for neurodegenerative diseases relies on reducing the innate tumorigenicity of pluripotent stem cells. However, the heterogeneity within iNSCs is a major hurdle in quality control prior to clinical applications. Herein, we generated iNSCs from human fibroblasts, by transfecting transcription factors using Sendai virus particles, and characterized the expression of iNSC markers. Using immunostaining and quantitative real time –polymerase chain reaction (RT –qPCR), no differences were observed between colonies of iNSCs and iNSC-derived neurons. Unexpectedly, patch-clamp analysis of iNSC-derived neurons revealed distinctive action potential firing even within the same batch product. We performed single-cell RNA sequencing in fibroblasts, iNSCs, and iNSC-derived neurons to dissect their functional heterogeneity and identify cell fate regulators during direct reprogramming followed by neuronal differentiation. Pseudotime trajectory analysis revealed distinct cell types depending on their gene expression profiles. Differential gene expression analysis showed distinct *NEUROG1*, *PEG3*, and *STMN2* expression patterns in iNSCs and iNSC-derived neurons. Taken together, we recommend performing a predictable functional assessment with appropriate surrogate markers to ensure the quality control of iNSCs and their differentiated neurons, particularly before cell banking for regenerative cell therapy.

## 1. Introduction

Neurodegenerative diseases, including Parkinson’s and Alzheimer’s disease, affect over 55 million people worldwide; their incidence increases as life expectancy increases [1,2,3]. Neurodegeneration is associated with a dysfunction of the synapse and neural networks, resulting in the loss of neuronal function. Recently, it has been reported that the self-renewal and differentiation abilities of neural stem cells (NSCs) gradually decrease with age [4], resulting in a decreased pool of NSCs over time [5]. Therefore, to relieve the progressive loss of the structure or function of neurons in neurodegenerative diseases, NSC-based treatments have emerged as an innovative approach to neuron regeneration [6,7,8].

NSCs self-renew into multipotent cells that give rise to neurons, astrocytes, and oligodendrocytes in the central nervous system. By detouring the pluripotency of embryonic stem cells (ESCs) or induced pluripotent stem cells (iPSCs) [9,10], directly reprogrammed NSCs (iNSCs) derived from somatic cells are relatively safe against the potential tumorigenic risk of stem cells [11,12]. However, owing to the heterogeneity of iNSCs, the quality control of iNSCs and their differentiated neurons during the biomanufacturing process remains a challenging issue for clinical applications. Although heterogeneity within NSCs has been reported [13,14,15,16], how this heterogeneity arises and affects the function of the differentiated neurons has not been fully elucidated.

In this study, we generated iNSCs by directly reprogramming human fibroblasts and differentiating them into neurons. We demonstrated that iNSC-derived neurons have distinct action potential firing depending on the colonies of iNSCs, even though the same batch has iNSC-specific characteristics. By further dissecting the functional heterogeneity within iNSC-derived neurons, using single-cell RNA sequencing (scRNA-seq), we found important surrogate markers reflecting the electrophysiological properties of iNSC-derived neurons. These could be very useful for the quality control of iNSCs or their differentiated neurons during the biomanufacturing of stem cell-based neuroregenerative medicine products.

## 2. Materials and Methods

### 2.1. Cell Culture

Human fibroblasts (CRL2097, ATCC) were cultured in minimum essential medium (MEM) supplemented with 10% fetal bovine serum (FBS), 1% MEM non-essential amino acids solution, and 1% penicillin-streptomycin at 37 °C in the presence of 5% CO_2_. All cell culture materials were purchased from Thermo Fisher Scientific, Waltham, MA, USA unless stated otherwise.

### 2.2. Direct Reprogramming of Human Fibroblasts into Neural Stem Cells

NSCs were generated by directly reprogramming human fibroblasts using previously reported protocols [13]. Briefly, human fibroblasts (3 × 10^4^ cells/mL) were incubated in fibroblast growth media containing CytoTune™-iPS 2.0 Sendai Reprogramming kit, carrying hKOS, hc-Myc, and hKlf4 (Cat# A16517, Thermo Fisher Scientific, Waltham, MA, USA). One day after infection, the Sendai virus (SeV) was eliminated, using direct reprogramming media supplemented with A83-01 (Tocris Bioscience, Bristol, UK), CHIR99021 (Tocris Bioscience), sodium butyrate (Sigma, St. Louis, MO, USA), 2-phospho-L-ascorbic-acid (Sigma), and recombinant human leukemia inhibitory factor (hLIF) (Peprotech, Rocky Hill, NJ, USA). The basal medium consisted of Neurobasal medium (Gibco, Carlsbad, CA, USA) and Advanced Dulbecco’s Modified Eagle Medium/Ham’s F-12 (DMEM/F12) (Gibco) supplemented with N-2 Supplement (Thermo Fisher Scientific), B27 Supplement, minus vitamin A (Gibco), Albumax-I (Gibco), Glutamax-1 (Gibco), and β-mercaptoethanol (Gibco). Seven days after starting reprogramming, the cells were dissociated with Accutase (Millipore, Burlington, MA, USA), containing Y-27632 dihydrochloride (Tocris Bioscience), and seeded on 6-well plates coated with Geltrex^™^ LDEV-Free Reduced Growth Factor Basement Membrane (Geltrex, Gibco). Between days 19 and 25, iNSC colonies were isolated and transferred to a Geltrex-coated 4-well plate filled with maintenance media consisting of supplements A83-01, CHIR99021, and hLIF in basal media. For temperature-sensitive SeV inactivation, iNSCs were cultured at 39 °C for 15 days. All media were changed every 2 days, and the cells were subcultured every 4–5 days. The iNSCs collected at passage 4 were cryopreserved in a maintenance media containing 10% DMSO, which was used as a cell-freezing medium.

### 2.3. Quantitative Real Time-Polymerase Chain Reaction (RT-qPCR)

Total RNA from human fibroblasts, commercially available iPSCs, and four colonies of directly reprogrammed iNSCs were extracted using an RNeasy Plus Mini Kit (QIAGEN, Hilden, Germany) according to the manufacturer’s instructions. After quantifying the isolated RNA, complementary DNA (cDNA) was synthesized using 1 µg/µL RNA and an iScript™ cDNA Synthesis Kit (Bio-Rad Laboratories, Hercules, CA, USA) according to the manufacturer’s instructions. RT-qPCR was performed with the cDNA template from each cell line and a QuantiTect SYBR Green PCR Kit (QIAGEN) using a 7900HT Fast Real-Time PCR System (Applied Biosystems, Waltham, MA, USA). The primer sequences used for the target genes (*COL1A1*, *PAX6*, *CDH2*, *POU5F1*, and *NANOG*) are listed in Appendix A. The fold changes in the genes were calculated using the 2^−△△Ct^ method with Expression Suite Software Version 1.2 (Thermo Fisher Scientific). Gene expression values were normalized to those of *GAPDH*, a housekeeping gene.

### 2.4. Immunocytochemistry

Cells were grown on Geltrex-coated four-well plates and fixed with 4% paraformaldehyde (PFA) (pH 7.4) for 10 min at room temperature (RT). After permeabilizing the cells with 0.5% Triton X-100 in Dulbecco’s phosphate-buffered saline (DPBS) at RT for 15 min, the cells were blocked with 3% bovine serum albumin (BSA) in DPBS at RT for 1 h. For the immunostaining of iNSCs or iPSCs, the cells were incubated at 4 °C overnight with the following primary antibodies: SOX1 (1:50, Thermo Fisher Scientific), PAX6 (1:50, Thermo Fisher Scientific), Nestin (1:50, Thermo Fisher Scientific), SOX2 (1:50, Thermo Fisher Scientific), Cadherin-2 (1:500, Cell Signaling Technology, Danvers, MA, USA), BLBP (brain lipid-binding protein, 1:200, Millipore), Ki-67 (1:500, Santa Cruz Biotechnology, Dallas, TX, USA), and NANOG (homeobox protein, 1:200, Cosmobio, Koto, Tokyo, Japan). For the immunostaining of differentiated neurons, the cells were incubated with primary antibodies against Tuj1 (tubulin beta-3, 1:200, Abcam, Cambridge, UK) and MAP2 (microtubule-associated protein 2, 1:200, Osenses Adelaide, SA, Australia). All primary antibodies were diluted with 0.3% BSA in DPBS. After washing with 0.1% BSA, the cells were incubated at RT for 1 h with the following secondary antibodies: Alexa Fluor-488-conjugated donkey anti-mouse (1:250, Thermo Fisher Scientific), Alexa Fluor-488-conjugated donkey anti-goat (1:250, Thermo Fisher Scientific), Alexa Fluor-555-conjugated donkey anti-rabbit (1:250, Thermo Fisher Scientific), Alexa Fluor-488-conjugated goat anti-mouse (1:500, Cell Signaling Technology), and Alexa Fluor-555-conjugated donkey anti-rabbit antibodies (1:250, Thermo Fisher Scientific). All secondary antibodies were diluted with 1% BSA in DPBS before use. The nuclei were then stained with DAPI (NucBlue™ Fixed Cell ReadyProbes Reagent, Thermo Fisher Scientific). The images of stained cells were acquired using a fluorescence microscope (Axio Observer 7, Carl Zeiss, Jena, Germany).

### 2.5. Karyotyping of Induced Neural Stem Cells (iNSCs)

Next, iNSCs were cultured in a Geltrex-coated T-25 flask with growth media to reach 50–60% cell confluency. For karyotyping analysis, 500 µL of KaryoMAX™ Colcemid^™^ Solution (Gibco) was added to each flask and incubated for 1 h. After washing with DPBS, the cells were collected using Accutase and centrifuged. The cell pellets were collected and reconstituted with 5 mL of 0.075 M potassium chloride solution and incubated at 37 °C for 25 min. After fixing cells in 500 µL Carnoy’s fixative solution (3:1 methanol/acetic acid), the supernatant was removed via centrifugation. This step was repeated thrice. The cell pellet was then placed on a glass slide and incubated at 60 °C for 30 min. The cells on the slide were treated with 50% H_2_O_2_ at RT for 3 min, followed by incubation at 60 °C for 30 min. Giemsa staining was used to analyze the karyotypes of the cells. Karyotyping analysis of iNSCs was performed by chromosome image processing system (ChIPS-Karyo) (Gendix Inc., Seoul, Republic of Korea) 

### 2.6. Short Tandem Repeat (STR) Profiling Analysis

To examine the genetic stability of iNSCs, DNA was extracted and quantified using a QIAamp DNA Mini Kit (QIAGEN) and Quantifier Trio DNA Quantification Kit (Thermo Fisher Scientific). Amplification was performed with 1 ng/µL of iNSC DNA, negative control (amplified grade water), and positive control (2800M DNA) (Promega, Madison, WI, USA), using the 5X Master Mix and 5X Primer Pair Mix in PowerPlex Kit (Promega) and Verti^®^ 96-well Thermal Cycler (Thermo Fisher Scientific). Capillary electrophoresis (CE)-based STR was performed using an Alleic Ladder Mix and WEN Internal Lane Standard 500 in a PowerPlex Kit and Genetic Analyzer 3500 (Applied Biosystems). The STR results were analyzed using GeneMapper ID-X software version 1.6.

### 2.7. Differentiation of Induced Neural Stem Cells (iNSCs) into Neurons

Here, iNSCs (2 × 10^5^ cells/mL) were seeded on a Geltrex-coated 4-well plate and cultured with growth media overnight at 37 °C in 5% CO_2_. The spent medium was changed to neuron differentiation media containing 20 ng/mL Brain Derived Neurotrophic Factor (BDNF) (Peprotech), 20 ng/mL Glial Derived Neurotrophic Factor (GDNF) (Peprotech), 0.5 mM dibutyryl cAMP (Enzo Life Sciences, Farmingdale, NY, USA), 50 µg/mL 2-phospho-L-ascorbic acid (Sigma), 10 µM γ-secretase inhibitor XXI, and Compound E (Millipore) to basal media consisting of DMEM/F12 (Gibco), B27 minus vitamin A (Gibco) and penicillin–streptomycin (Gibco). To induce their differentiation into neurons, iNSCs were maintained and cultured for 20 days, and half of the spent medium was replaced every 2–3 days. Compound E, a differentiation promoter, was added to the medium for the first 10 days.

### 2.8. Electrophysiology Assays for the Differentiated Neurons

Whole-cell patch-clamp recordings were performed to measure action potentials (APs) and voltage-gated sodium currents. iNSCs (2 × 10^5^ cells/mL) were seeded on 12 mm Geltrex-coated cover glasses, which were placed into each well of a 4-well plate, and differentiated into neurons over 3 weeks in a culture incubator at 37 °C with 5% CO_2_. For whole-cell patch-clamp recordings, cover glasses with cells were carefully placed in the recording chamber in external solutions consisting of 137 mM NaCl, 10 mM HEPES (4-(2-Hydroxyethyl)piperazine-1-ethanesulfonic acid), 20 mM glucose, and 2 mM CaCl_2_ (adjusted to pH 7.3 with NaOH, 290–300 Osm). The internal solutions containing 140 mM K-gluconate, 20 mM KCl, 5 mM NaCl, 0.5 mM EGTA (Ethylene bis(oxyethylenenitrilo)tetraacetic acid tetrasodium), 1 mM MgCl_2_, and 10 mM HEPES (pH 7.25, 280–290 Osm) were put into a glass micropipette. All the chemicals used in the electrophysiological solutions were purchased from Sigma-Aldrich (St. Louis, MO, USA). Patch pipettes were made from borosilicate glass capillaries (Harvard Apparatus, Holliston, MA, USA) using a micropipette puller (Narishige Scientific Instrument Laboratory, #PC-100, Tokyo, Japan). The pipette resistance was 5–8 MΩ when filled with the internal solution. Spontaneous APs were measured in gap-free mode. Evoked APs were induced by injections of step currents from −0.1 to 0.2 nA in 0.02 nA increments for 500 ms. For the voltage-gated Na^+^ currents, we used voltage steps for 1 s from −70 to +70 mV in 10 mV increments. Following the application of voltage steps and the recording of inward currents, 100 nM tetrodotoxin (TTX) was administered to selectively inhibit TTX-sensitive inward currents, enabling the specific analysis of Na^+^ currents. Signals were filtered at 5 kHz and sampled at 10 kHz using an Axopatch 200 B amplifier (Molecular Devices, Sunnyvale, CA, USA). Voltage- or current-clamp protocol generation and data acquisition were controlled using computers equipped with an A/D converter (Digidata 1550 (Molecular Devices) and pClamp 11.3 software (Molecular Devices).

### 2.9. Single-Cell RNA Sequencing

#### 2.9.1. Single-Cell Preparation

The cells were detached using Accutase and washed with serum-containing DMEM, followed by washing with cold BSA/PBS twice. After counting the cells using acridine orange/propidium iodide staining (Logos Biosystems, Anyang-si, Gyeonggi-do, Republic of Korea, Cat# F23001) with a LUNA-FX7 ^TM^ Automated Fluorescence Cell Counter (Logos Biosystems), the samples were tagged with antibody-polyadenylated DNA barcodes for human cells (BD Biosciences, Franklin Lakes, NJ, USA, Cat# 633781).

#### 2.9.2. Library Preparation and Single-Cell RNA Sequencing

Single cells were captured using a BD Rhapsody Express instrument (BD Biosciences) according to the manufacturer’s instructions. Briefly, pooled cells were loaded into a BD Rhapsody cartridge (Cat #633731; BD Biosciences). After separating the cells, magnetic beads with cell barcodes (cell capture beads) were loaded into the cartridge. The mRNA from the lysed cells was captured using the beads. cDNA synthesis and exonuclease I treatment were performed on the mRNA-capture beads using a BD Rhapsody cDNA kit (BD, Cat# 633773). Following the manufacturer’s protocols for mRNA whole transcriptome analysis (WTA) and Sample Tag Library Preparation (BD Biosciences), scRNA-seq libraries were constructed using a BD Rhapsody WTA amplification kit (BD Biosciences, Cat# 633801). The cDNA was sequentially subjected to random priming extension (RPE), RPE amplification, and index PCR. Next, for the sample tag library, cDNA was sequentially subjected to nested and index PCR. Purified WTA and sample tag libraries were quantified using qPCR, according to the qPCR Quantification Protocol Guide (KAPA), and qualified using an Agilent Technologies 4200 TapeStation (Agilent Technologies, Santa Clara, CA, USA). After the libraries were pooled, 150-bp paired-end reads were generated via sequencing using the HiSeq platform (Illumina, San Diego, CA, USA).

### 2.10. Data Processing

Raw sequencing data were processed using the BD Rhapsody WTA Analysis Pipeline v 1.11 (BD Biosciences, Franklin Lakes, NJ, USA) with GRCh39 (human) as the reference data, which was downloaded from the Ensembl database.

#### 2.10.1. Clustering Analysis

We classified the cells into transcriptionally similar clusters to identify distinctive genes depending on cell type. We filtered 64,185 cells based on quality control standards, with a fraction of counts from mitochondrial genes per cell. To remove the accounted injured cells, cells with a mitochondrial gene expression percentage of more than 20% were excluded. Downstream analysis was performed using the R (v. 4.0.3) package Seurat (v. 4.3.0). Briefly, we normalized the gene expression data using the SCTransform function. To adjust for within-batch variations, seven datasets were integrated using the functions, SelectIntegrationFeatures, PrepSCTIntegration, FindIntegrationAnchors, and IntegrationData. The number of principal components used for graphical clustering was determined using the RunPCA function. Ten clusters were visualized on a uniform manifold approximation and projection (UMAP) using the FindNeighbors, FindClusters, and RunUMAP functions.

#### 2.10.2. Analysis of Differentially Expressed Genes

Differentially expressed genes (DEGs) were analyzed between two groups, (nAP)-iNSCs vs. (AP)-iNSCs and nAP Neurons vs. AP Neurons, in clusters of interest using the function of R package Seurat, FindMarkers. The results of the DEG analysis are displayed as volcano plots.

#### 2.10.3. Pseudotime Trajectory Analysis

The R package Monocle3 was used to construct a branched pseudotime trajectory. Pseudotime plots of the iNSC and neuron groups were created for the expression of the genes of interest.

#### 2.10.4. Gene Ontology Analysis

Gene ontology (GO) analysis was performed using the ClusterProfiler (v. 4.6.2) package in R. The ClusterProfiler package also supports the statistical analysis and visualization of functional profiles for genes and gene clusters. Functional profiling of biological processes was also conducted using the DEGs obtained from early data processing.

### 2.11. Statistical Analysis

RT-qPCR data were analyzed using unpaired Student’s *t*-tests and presented as the mean ± standard error of the mean (SEM). Statistical analyses were performed using GraphPad Prism software (v. 9.5.1). For scRNA-seq statistics data, values provided by the R package ClusterProfilerwere used. (* *p* < 0.05, ** *p* < 0.005, *** *p* < 0.001).

## 3. Results

### 3.1. Direct Reprogramming and Characterization of Induced Neural Stem Cells (iNSCs)

iNSCs were successfully generated by directly reprogramming human fibroblasts via transfection of transcription factors (*OCT4*, *SOX2*, *c-Myc*, and *Klf4*) using Sendai virus particles. Morphological changes in the cells were observed 3 days after transfection, followed by colony formation on day 7. In particular, rosette morphology, an iNSC-specific shape, was observed after 10 days of direct reprogramming. Colonies of iNSCs were picked between days 19 and 25 for further expansion (Figure 1A). An inactivation process was used to eliminate the risk of residual virus in the final target cells. Sendai virus is known to be sensitive to temperature [17], and we found that it was inactivated after 15 days of culture. Inactivation of the Sendai virus in the final product of iNSCs was confirmed via RT-qPCR, where viral gene expression was not detected.

To characterize the directly reprogrammed iNSCs, we examined the mRNA and protein expressions of cell type-specific markers using RT-qPCR and immunocytochemistry analysis, respectively. We divided iNSCs into two groups depending on the action potential firing of their differentiated neurons; (nAP)-iNSCs were named after an iNSC colony whose differentiated neurons show no action potential (nAP Neuron). (AP)-iNSCs were named after iNSC colonies whose differentiated neurons showed action potentials (AP Neuron) (Appendix A). The expression of NSC-specific markers (*PAX6* and *CDH2*) was significantly increased in both the nAP and AP groups of iNSCs compared to those in fibroblasts or iPSCs (*n* = 3 per group, * *p* < 0.05) (Figure 1B). Interestingly, the expression of *PAX6* was significantly higher in (AP)-iNSCs than in (nAP)-iNSCs (*n* = 3 per group; * *p* < 0.05). The pluripotent stem cell markers *NANOG* and *POU5F1* were highly expressed in iPSCs, but not in fibroblasts or iNSCs. *COL1A1*, a fibroblast marker, was not detected in either (nAP)-iNSC or (AP)-iNSCs. The protein expression of NSC-specific marker genes was examined using immunocytochemistry. *NANOG*, a pluripotent stem cell marker, was not detected in directly reprogrammed iNSCs. In contrast, NSC-related protein markers (such as Pax-6, Nestin, SOX-1, SOX-2, Cadherin-2, and BLBP) were detected in both (nAP)- and (AP)-iNSCs. We found that the localization of Nestin expression was different between the two groups. (nAP)-iNSC showed a radial extending morphological feature and (AP)-iNSC showed short or non-observing processes (Figure 1C).

### 3.2. Evaluation of the Genetic Stability of Induced Neural Stem Cells (iNSCs)

The genetic stability of iNSCs from all colonies was confirmed via karyotyping analysis. The results showed a normal karyotype for all colonies of iNSCs in both the nAP and AP groups (Appendix A). Short tandem repeat (STR) profiling of iNSCs was well-matched for nine STR loci (Amelogenin, CSF1PO, D13S317, D16S539, D5S818, D7S820, TH01, TPOX, and vWA) in human fibroblasts, with a starting cell line provided by ATCC (Appendix A). From the results of the STR profiling analysis, no difference in the STR profiles of the 24 STR loci between (nAP)-iNSCs and (AP)-iNSCs was found (Appendix A.

### 3.3. Neuronal Differentiation Electrophysiology Assays

To verify the multipotency of the iNSCs, we further differentiated them into neurons (Figure 2A). After 7–10 days of differentiation, the neurites stretched out, followed by the appearance of axons and dendrites (Figure 2A, lower panel). Tubulin beta-3 chain (Tuj1) and microtubule-associated protein 2 (MAP2) were used as neuron-specific markers; their expression levels were detected via immunostaining (Figure 2B).

For the functional evaluation of iNSC-derived neurons, we measured the APs using the whole-cell patch clamping method with or without electrical stimulation. Unexpectedly, the firing rate of mature APs varied among the 27 iNSC-derived neuron samples (0%, 32.0~66.7% of the cells), even though morphologically verified neurons differentiated from the same batch of iNSC colonies. For further verification, we divided the iNSC-derived neurons into two groups, depending on the presence of AP firing (Figure 2C). nAP Neurons are iNSC-derived neurons with no spontaneous AP firing, whereas AP Neurons are designated iNSC-derived neurons with action potential firing with or without electrical stimulation (Figure 2C). Typical neuronal action potentials, in which firing increased in proportion to the magnitude of the current of electrical stimulation, were observed in AP Neurons (Figure 2D).

To examine the sodium ion channel current of AP Neurons, a 100 nM TTX, AP Neurons were treated with a sodium channel inhibitor; the results showed complete suppression of the current (Appendix A). Although the characterization of iNSCs using RT-qPCR and immunostaining revealed the NSC-specific properties of iNSCs for all colonies, the electrophysiological properties of each iNSC-derived neuron were not the same, suggesting heterogeneity in iNSC colonies as a source cell for differentiation.

### 3.4. Single-Cell Profiling of Directly Reprogrammed Induced Neural Stem Cells and Induced Neural Stem Cell-Derived Neurons

To investigate the distinct electrophysiological properties of iNSC-derived neurons, we performed scRNA-seq analysis of iNSC-derived neurons, iNSCs, and fibroblasts. To track the originally shifted gene expression in iNSCs depending on the different colonies, we used five groups of samples for scRNA-seq analysis: fibroblasts as raw material, (nAP)-iNSCs, (AP)-iNSCs, nAP Neurons, and AP Neurons.

We identified 64,185 cells through BD Rhapsody WTA Analysis Pipeline (v. 1.11). Unhealthy cells expressing more than 20% of mitochondrial genes (Appendix A) were excluded in following data analyses. After quality control, 59,140 cells were plotted on a UMAP with 10 clusters and used for subsequent analysis (Figure 3A and Appendix A). The numbers of DEGs from DE analyses of AP group vs. nAP group were compared. The data revealed that the number of downregulated DEGs was greater than the upregulated DEGs in both iNSCs and iNSC-derived neurons (Appendix A). Total UMAP projections from all samples were segregated by sample groups: iNSCs vs. neurons and nAP groups vs. AP groups (Figure 3B). The cluster proportions of each UMAP-projected sample are depicted with bar graphs; the proportion of cluster 2 was prominently increased in AP Neurons compared to the other groups. The proportion of cluster 10 was increased in iNSCs and maintained in nAP Neurons, but not in AP Neurons (Figure 3C). The GO analysis of biological processes (BP) in clusters 2 and 10 is shown in Figure 3D. Compared to (nAP)-iNSCs, the anterior/posterior pattern specification process was significantly decreased in both cluster 2 and 10 in the (AP)-iNSC group, while dense core granule localization and neuron fate commitment were increased in cluster 10 in (AP)-iNSCs.

In iNSC-derived neurons, the regulation of membrane potential and neuron projection development were upregulated in the AP Neuron group, reflecting their functional maturation into neurons, including action potential firing. In contrast, the stem cell population maintenance and cell cycle-related genes were downregulated in the AP SNeuron group, compared to the nAP Neuron group.

### 3.5. Molecular Characterization of Induced Neural Stem Cells and Induced Neural Stem Cell-Derived Neurons Using Single-Cell RNA-seq

Based on the cell type-related characterization of the genes in each cluster, we annotated all 10 clusters into four cell types: fibroblasts (clusters 1, 5, 7, and 8), iNSCs (cluster 3), iPSCs (cluster 4), and neurons (clusters 2, 6, 9, and 10) (Figure 4A). The proportions of the annotated cell types in the five scRNA-seq sample groups are shown in Figure 4B. Remarkably, in the AP Neuron group, the proportion of neuron-related cells was more than 80%, whereas the proportion of iNSC/iPSC-related cells was far less than that of other cell types, suggesting that the (AP)-iNSCs were successfully differentiated into AP Neurons. The feature plots of the cell type-specific gene expression levels also demonstrated proper annotation for each cell type: *COL1A1* (fibroblast group), *PAX6* and *NES* (iNSC groups), and *MAP2* (neuron groups) (Appendix A). In contrast, iNSC-derived glial cells were barely expressed in all groups from the feature plot for glial cell-specific genes (*AIF1* for microglia, *GFAP* for astrocytes, and *OLIG2* for oligodendrocytes) (Figure 4C), demonstrating that the iNSCs differentiated into neurons, but not glial cells, under our differentiation protocol. Using all samples in the direct reprogramming of fibroblasts into iNSCs and further differentiation into neurons, we created a pseudotime trajectory plot on UMAP, which also consistently reflected the experimental processes (Figure 4D). When iNSC or neuron-specific genes were assigned pseudotime, the expression of *PAX6*, a neural stem cell-specific marker, transiently increased, with a peak in iNSC-annotated cells in iNSC samples (Figure 4E left panel, Appendix A). In contrast, the expression of *MAP2*, a neuronal marker, gradually increased and peaked in neuron-annotated cells in the iNSC-derived neuron samples (Figure 4E, right panel).

### 3.6. Discovering Potential Surrogate Markers of Induced Neural Stem Cells from Functional Neurons

To filter out candidate surrogate markers for predicting the functional heterogeneity of iNSC-derived neurons, we chose DEGs with AP/nAP ratios, |log2FC| ≥ 1, and adjusted *p*-value < 0.05 from clusters 2 and 10 (Figure 5A). We selected three common significant DEGs in both iNSC and iNSC-derived neurons for each cluster, including *HOXB9* (homeobox B9), *NEUROG1* (neurogenin-1), and *HOXB7* (homeobox B7) in cluster 2 and *ARMCX1* (Armadillo repeat-containing X-linked 1), *STMN2* (Stathmin-2), and *PEG3* (paternally expressed 3) in cluster 10. Among these six genes, we selected the top three genes having the highest proportion of cells expressing them: *STMN2* (adjusted *p*-value < 0.05), *NEUROG1* (adjusted *p*-value < 0.05), and *PEG3* (adjusted *p*-value < 0.05) (in that order) (Figure 5B). To use these genes as potential surrogate markers, we visualized the DEG results from the total clusters as a violin plot. The fold changes in *NEUROG1* and *PEG3* expression based on the AP/nAP-iNSC ratio decreased to 0.12 (adjusted *p*-value < 0.05) and 0.26 (adjusted *p*-value < 0.05), respectively. For *PEG3* and *STMN2*, their fold changes based on the AP/nAP Neuron ratio increased to 1.62 (adjusted *p*-value < 0.05) and 2.43 (adjusted *p*-value < 0.05), respectively. Interestingly, the fold change in *NEUROG1* expression in AP Neurons decreased to 0.06 (adjusted *p*-value < 0.05) (Figure 5C). Based on these results, we suggest that *NEUROG1*, *PEG3*, and *STMN2* are potential surrogate markers for determining the functional heterogeneity of iNSCs and iNSC-derived neurons.

## 4. Discussion

Neural stem cell heterogeneity has been reported in mice [13,16] and humans [15]. To address the heterogeneity of iNSCs and the neurons differentiated from them, we first successfully generated iNSCs by directly reprogramming human fibroblasts using Yamanaka factors. Unlike conventional methods that use potentially mutagenic viruses, such as lentiviruses and retroviruses, for gene transfer in direct reprogramming, we utilized the Sendai virus, because of its non-integration, low cytotoxicity, and high efficiency [18]. The genetic stability of iNSCs from each colony was confirmed via STR profiling and karyotyping. All batches of directly reprogrammed iNSCs showed neural stem cell-specific properties based on microscopic morphological observation, RT-qPCR, and immunohistochemistry. In addition, the expression of neural stem cell markers such as Pax6, Cadherin, Nestin, SOX-1, SOX-2, and BLBP [13,19] increased in iNSCs from all colonies compared to those in human fibroblasts, confirming the successful direct reprogramming of fibroblasts into NSCs. However, we observed that the AP firing of differentiated neurons (iNSC-derived neurons) varied depending on the iNSC colony, although there was an increase in the expression of neuronal markers (Tuj1 and MAP2) in all iNSC-derived neurons. To the best of our knowledge, this is the first study to demonstrate the functional heterogeneity of iNSC-derived neurons depending on the colony of iNSCs used.

The functional heterogeneity of iNSC-derived neurons was further dissected using the scRNA-seq of two groups, based on the absence or presence of action potential firing, termed the nAP and AP groups, respectively. Retrospectively, iNSCs were divided into two groups according to their differentiated neurons’ action potential: (nAP)-iNSCs and (AP)-iNSCs. Interestingly, we found a significant difference in the expression of PAX6 between (nAP)-iNSCs and (AP)-iNSCs using RT-qPCR. PAX6-positive iNSCs differentiate into functional neurons with APs in mice [13]. From immunocytochemistry analysis, we found a difference in Nestin localization between (nAP)-iNSCs and (AP)-iNSCs. There are two types of Nestin-positive cells with different localizations in the mouse hippocampal dentate gyrus, depending on the differentiation stage of neurogenesis [20]. In this study, we propose that the characterization of Nestin localization in the neural stem cell can contribute to the quality control of the heterogeneity of iNSCs. To investigate whether gene expression changes in iNSCs can affect the action potential firing of neurons differentiated from them, we further analyzed the significant DEGs in each cluster of the AP and nAP groups using scRNA-seq data.

Our scRNA-seq results showed that expression of *NEUROG1* decreased in both AP-iNSC and AP Neurons. *NEUROG1* is transiently expressed during embryonic development [21] and suppresses neuronal differentiation [22]. These previous reports are consistent with our scRNA-seq results, which showed lower *NEUROG1* expression in the AP group than in the nAP group.

Another surrogate marker, *PEG3*, was significantly upregulated in AP Neurons, while its expression decreased in (AP)-iNSCs. Previous studies showed that PEG3 knockdown increases the expression of pluripotency-related genes in mouse ESCs [23]. *PEG3* is also highly expressed in neuronal cells, protecting against excessive neuronal loss by inhibiting caspase 3-mediated apoptosis [24]. Although we suggest *PEG3* as a surrogate marker, we cannot rule out the possibility that other sex chromosome-related genes, such as MEG3 [25], have been identified using female-originated cells.

Finally, *STMN2* expression was increased in AP Neurons. A deficiency in *STMN2* causes amyotrophic lateral sclerosis (ALS). STMN2 was downregulated in an in vitro ALS model using the mutant TDP-43 (TAR DNA-binding protein-43) cell line. Preventing STMN2 degradation increases neurite extension, suggesting the important role of *STMN2* in regulating neurodegeneration [26,27]. Neurite growth is closely related to AP firing in several ways, such as developing APs [28] or stopping growth by AP firing [29]. Therefore, *STMN2* is suggested as a potential surrogate marker for predicting the heterogeneity of directly reprogrammed cells, as well as a regulatory gene for AP firing.

In this study, we identified several potential markers in iNSCs and iNSC-derived neurons for predicting functional heterogeneity using scRNA-seq and unsupervised analysis (Figure 6). However, our study has several limitations. Like numerous other studies, we conducted electrophysiological assessments following direct cell reprogramming, yielding successful results for spontaneous and stimulated APs [30,31]. Despite obtaining AP measurements from directly reprogrammed and differentiated neurons, further investigations using whole-cell patch-clamp recordings have been conducted to discern the ionic properties of these differentiated neurons, including the measurement of inhibitory or excitatory postsynaptic currents [32,33]. Additionally, other studies involved the immunostaining of differentiated neurons, revealing that directly reprogrammed and differentiated neurons expressed various markers, such as dopaminergic, glutamatergic, and GABAergic markers, among others [13]. However, characterizing these directly reprogrammed iNSCs for potential clinical applications remains challenging.

## Figures and Tables

**Figure 1 cells-12-02818-f001:**
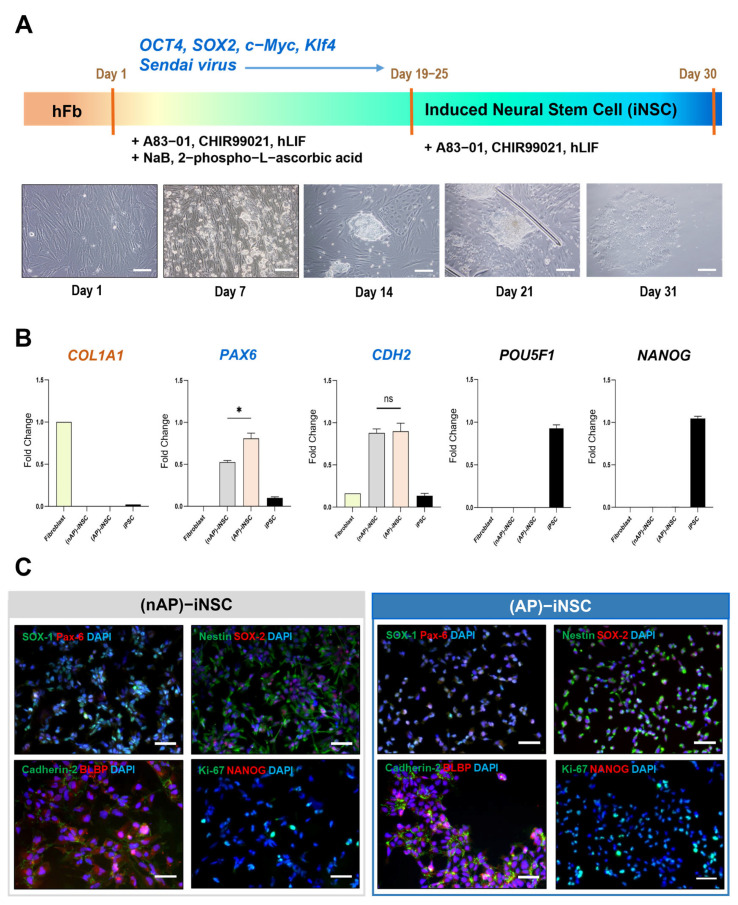
Characterization of directly reprogrammed neural stem cells. (**A**) Schematic diagram of the direct reprogramming of human fibroblasts (hFbs) into neural stem cells (iNSCs). Human transcription factors (*OCT4*, *SOX2*, *c-Myc*, and *Klf4*) were transfected into hFbs using Sendai virus particles. Representative phase contrast micrographs of directly reprogrammed cells are shown in the lower panel. Scale bar = 200 µm. (**B**) RT-qPCR analysis of cell-specific markers after 30 days of direct reprogramming; *COL1A1* for fibroblasts, *PAX6* and *CDH2* for iNSCs, and *POU5F1* and *NANOG* for induced pluripotent stem cells (iPSCs). Data are presented as the mean ± standard error of the mean (SEM) of three independent samples per experiment. N = 3, * *p* < 0.05, paired Student’s *t*-test. Ns: not significant. (**C**) Immunocytochemistry of iNSC or iPSC-specific marker proteins at day 30 of direct reprogramming followed by re-suspension after iNSC colony picking. DAPI is counterstained for nucleic acid. Three independent samples per experiment. Scale bar = 50 µm. **SOX1**; SRY-box transcription factor 1, **SOX2**; SRY-box transcription factor 2, **PAX6**; paired box protein Pax6, **Nestin**; a cytoskeletal intermediate filament protein, **Cadherin-2**; N-Cadherin, **BLBP**; brain lipid-binding protein, **NANOG**; homeobox protein NANOG.

**Figure 2 cells-12-02818-f002:**
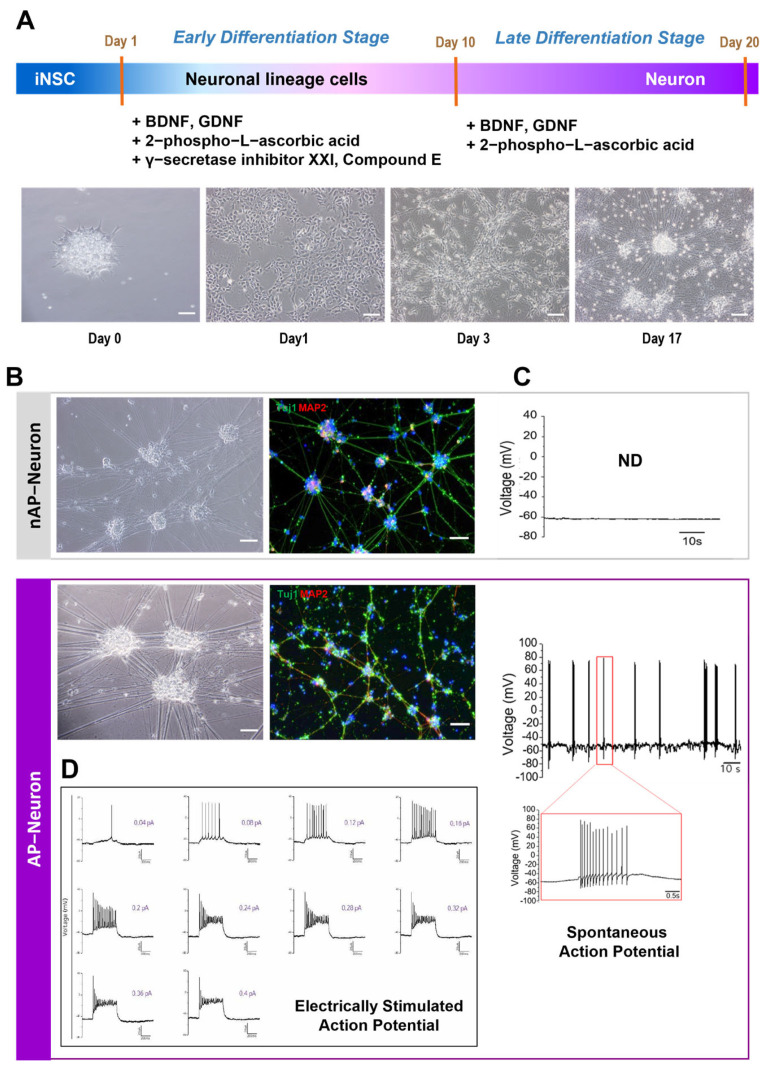
Differentiation of iNSCs into neurons and functional assessment of neurons. (**A**) Schematic diagram of differentiation of induced neural stem cells (iNSCs) into neurons. Representative phase contrast images of cells during differentiation are shown in the lower panel. Scale bars = 200 µm. (**B**) Immunostaining of neurons for specific marker proteins: tubulin beta-3 (**Tuj1**) and microtubule-associated protein 2 (**MAP2**). Three independent samples per experiment. Scale bar = 100 µm. (**C**) Whole-cell patch-clamp recordings of the action potentials in iNSC-derived neurons. Spontaneous action potentials were not detected (ND) in nAP Neurons but were observed in AP Neurons. AP Neurons showed a typical action potential firing trace. (**D**) Electrically stimulated action potentials in AP Neurons.

**Figure 3 cells-12-02818-f003:**
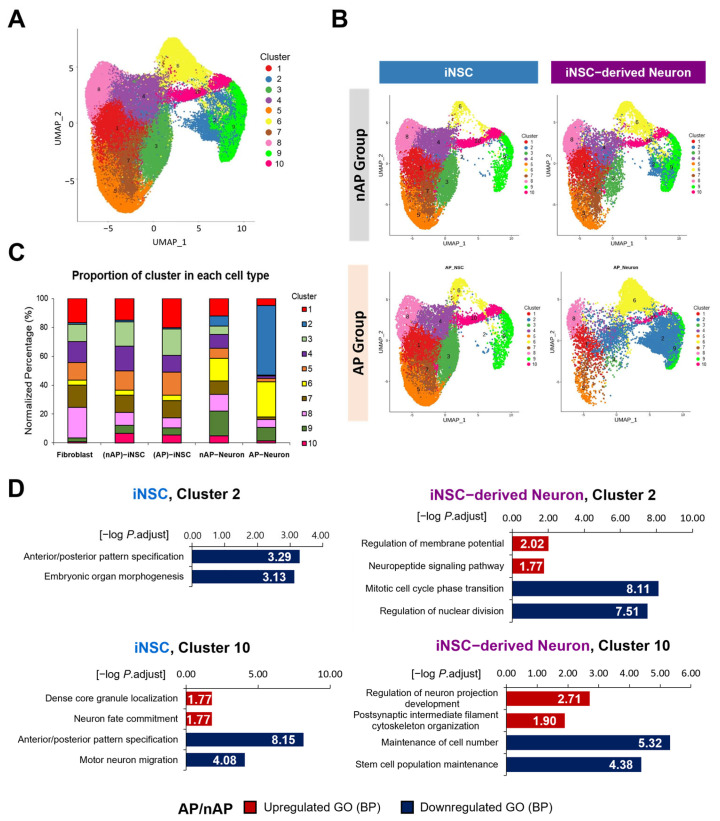
Identification of cell clusters in induced neural stem cells (iNSCs) and iNSC-derived neurons using scRNA-seq analysis. (**A**) Total filtered cells were projected on the UMAP and clustered into 10 groups with unsupervised clustering. This is the UMAP projection of 59,140 single cells expressing less than 20% of mitochondrial genes during and after direct reprogramming. Each dot represents one cell, with a color code for the cell types. (**B**) The proportion of clusters for each sample depends on the presence of action potential firing of neurons using an unsupervised method. Each UMAP of cells is allocated into the nAP or AP groups for iNSCs and their differentiated neurons. (**C**) The cluster proportion of each group sample is depicted on the bar graph with normalization to a maximum of 100%. (**D**) Gene ontology (GO) analysis of biological processes (BP) in clusters 2 and 10 in the AP group, compared to the nAP group of iNSCs or iNSC-derived neurons. BPs related to neuronal function are depicted on the bar graph.

**Figure 4 cells-12-02818-f004:**
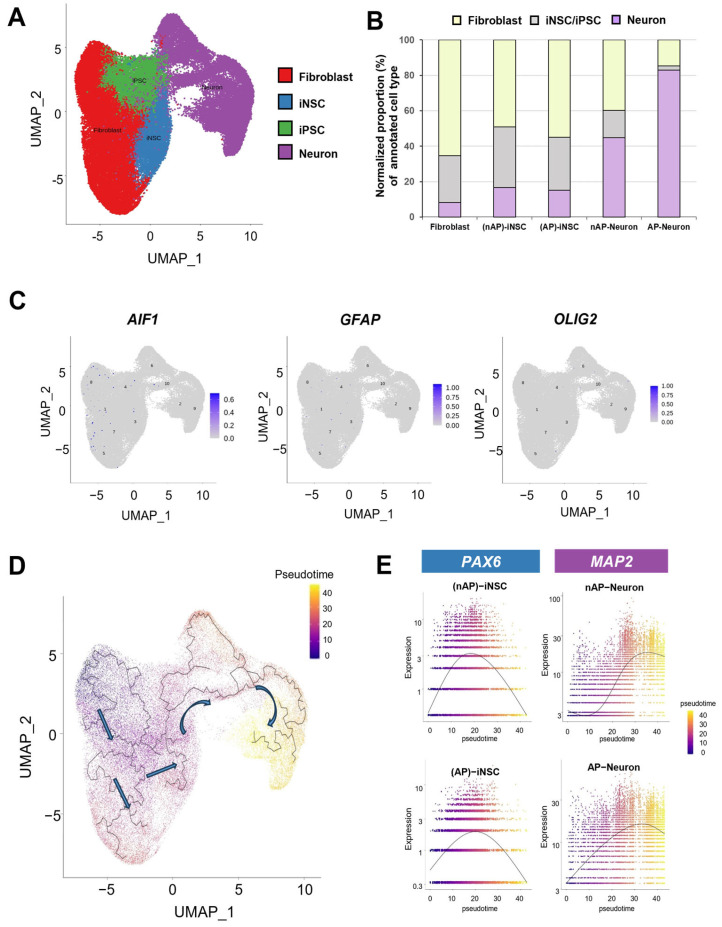
Pseudotime analysis of the direct reprogramming of fibroblasts to iNSCs and subsequent differentiation into neurons. (**A**) Projected UMAP being clustered by cell type-specific gene expression: fibroblast (red), iPSCs (green), iNSCs (blue), and neurons (purple). (**B**) The population of neurons gradually increased throughout the differentiation process. (**C**) Glial cell markers, such as *AIF1* (microglia), *GFAP* (astrocytes), and *Olig2* (oligodendrocytes), were rarely observed across all 10 clusters after direct reprogramming and differentiation. Numbers on the plots indicate clusters. (**D**) Pseudotime trajectory depicting the successful direct reprogramming of fibroblast into iNSCs and further differentiation into neurons. Arrows indicate the overall direction of cell differentiation based on pseudotime. (**E**) The expression level of PAX6, an iNSC-specific marker, was highly increased at the pseudotime of the direct reprogramming process for iNSCs and decreased while reaching the neuron-annotated regions. The expression of MAP2, a neuron-specific marker, was increased throughout the differentiation process into neurons.

**Figure 5 cells-12-02818-f005:**
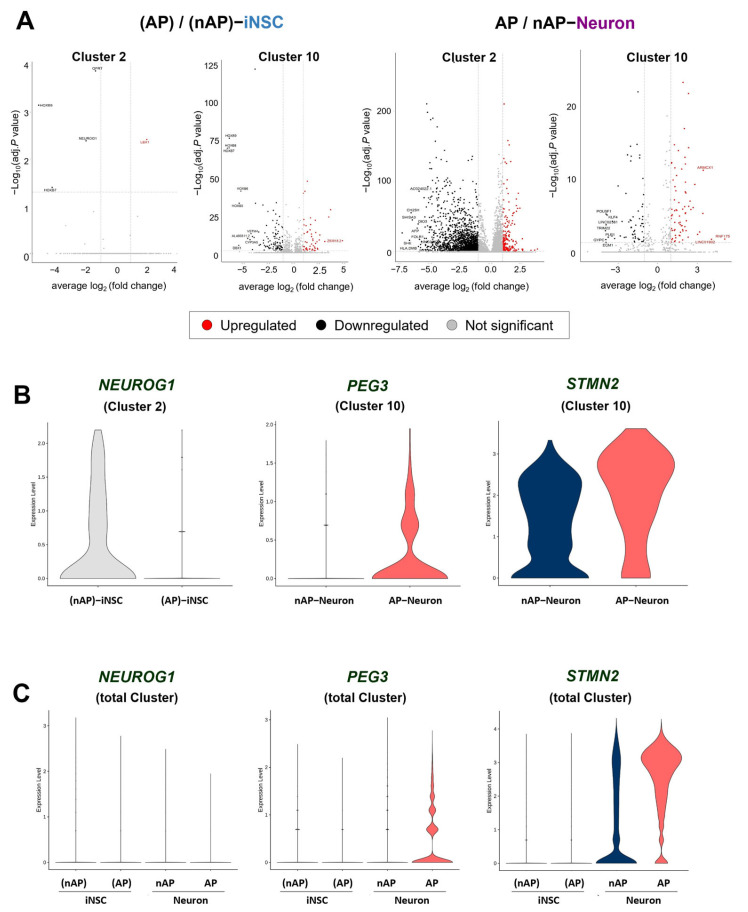
Discovering potential surrogate markers for functional iNSC-derived neurons. (**A**) Volcano plot for differentially expressed genes (DEGs) in (AP)-iNSCs compared to (nAP)-iNSCs from clusters 2 and 10 (left panel) and AP Neurons compared to nAP Neurons (right panel). The criteria for DEG significance is |log2FC| ≥ 1 and adjusted *p*-value ≤ 0.05 (gray dashed line). (**B**) Expression violin plot of *NEUROG1* (cluster 2), *PEG3* (cluster 10), and *STMN2* (cluster 10). (**C**) *NEUROG1*, *PEG3*, and *STMN2* gene expression from the total cluster groups are depicted using a violin plot.

**Figure 6 cells-12-02818-f006:**
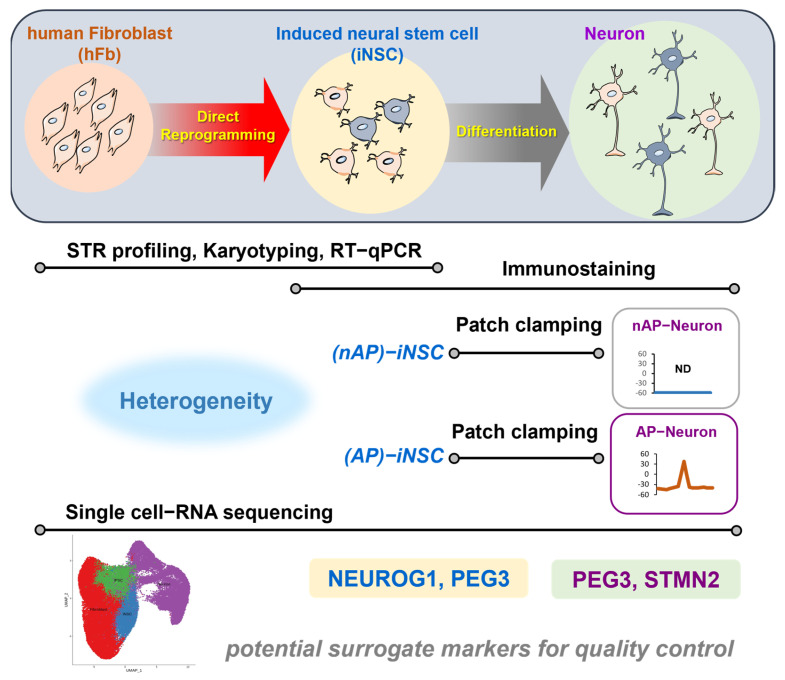
Graphical summary of the study. Potential surrogate markers (*NERUOG1, PEG3*, and *STMN2*) can be investigated via single-cell RNA sequencing for quality control and the prediction of functional heterogeneity of directly reprogrammed induced neural stem cells (iNSCs) and iNSC-derived neurons.

## Data Availability

The data that support the findings of this study are available from the corresponding author upon reasonable request.

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
