# Peer review of "Exploring the Functional Heterogeneity of Directly Reprogrammed Neural Stem Cell-Derived Neurons via Single-Cell RNA Sequencing"

_cells, 2023, doi:10.3390/cells12242818_

Round 1
Reviewer 1 Report
Comments and Suggestions for Authors
In this paper, authors have found functional assessment assay with appropriate surrogate markers to ensure the quality control of iNSCs and their differentiated neurons as they are very heterogeneous.
Minor comments:
1. In Results section 3.1 introduce what is AP and nAP group of iNSCs. Authors introduce the terms later, move it before.
2. Write the p values as well along with stating whether * or not.
3. In fig 1 B , Keep the PAX6 scale same as others.
4. In fig 1 legend, clearly define what 3 independent samples are..coverslips, experiments etc.
5. In fig 2B, how many coverslips or experiments were done?
6. In clustering analysis, what was the reason of choosing 10 clusters?
Author Response
We appreciate the valuable comments of the reviewer.
Overall comments: In this paper, authors have found functional assessment assay with appropriate surrogate markers to ensure the quality control of iNSCs and their differentiated neurons as they are very heterogeneous.
[Minor Comments]
Comment 1: In Results section 3.1 introduced what is AP and nAP group of iNSCs. Authors introduced the terms later, move it before.
Response 1: We added the introduction for the terms, AP and nAP, on page 6, lines 255–261.
Comment 2: Write the p values as well along with stating whether* or not.
Response 2: We added ‘ns’ in the Figure 1B, denoting “not significant” on page 7, lines 275–288. We wrote the adjusted p values for DEG data clearly in Figures 15B, 5C, and text on page 14, lines 412–420.
Comment 3: In Fig 1B, keep the PAX6 scale the same as others.
Response 3: We adjusted the scale for PAX6 from RT-qPCR results in Fig 1B, and kept it the same as for the others.
Comment 4: In Fig 1 legend, clearly define what 3 independent samples are coverslips, experiments etc.
Response 4: We clearly defined it as “three independent samples per experiment” on page 7, lines 275–288.
Comments 5: In Fig 2B, how many coverslips or experiments were done?
Response 5: For immunostaining, we used three independent samples. We stated this as “three independent samples per experiment” in the legend of Fig 2B on page 9, lines 305–313.
Comments 6: In clustering analysis, what was the reason of choosing 10 clusters?
Response 6: For clustering analysis, first, we used the Seurat package in R for the analysis. To determine the characteristics of clusters, we selected 12 principal components (PCs) from the elbow plot. This information was written in “Materials and Methods” on page 5, lines 216–222.
Second, graph-based clustering of the Seurat package was used, and we used the internal functions, FindNeighbors(), FindClusters(), and RunUMAP(), with 12 PCs as described on page 5, lines 219–222.

Reviewer 2 Report
Comments and Suggestions for Authors
In this manuscript, Kim et al. delve into the exploration of the heterogeneity present in neurons derived from directly reprogrammed neural stem cells, employing single-cell RNA sequencing. Their focus lies in comparing the distinctions between induced neural stem cells (iNSCs) and iNSC-derived neurons, revealing conspicuous heterogeneity within these cell types. The findings of this study align with existing conclusions emphasizing the prevalence of heterogeneity in directly reprogrammed neural stem cells. Regrettably, I am inclined to believe that this manuscript does not significantly contribute to our current knowledge, and therefore, I do not consider it a strong candidate for publication in Cells.
Here are specific comments:
I lack expertise in neuron biology, and I did not find a clear definition of "nAP-iNSC" and "AP-iNSC" in the manuscript (unless I overlooked it). Could you clarify if these terms refer to the presence of action potentials (AP) firing or spontaneous AP firing? Additionally, the reasons behind the emergence of two distinct cell groups following reprogramming are not adequately explained. Is there any clarification available?
Observing Figure 1C, it appears that the expression of Ki-67 and nestin is substantially higher in (AP)-iNSC compared to (nAP)-iNSC. The localization of these markers also seems different, despite their presence in both cell groups.
Is there any information regarding the relationship between AP-iNSC and AP-Neurons? According to the description in Lines 325-328, it is stated that AP-iNSC generates AP-Neurons, while nAP-iNSC generates nAP-Neurons. Could you elucidate the reasons for this distinction?
If I understand correctly, the significance of marker genes lies in predicting functional neurons, and the screening of these genes is based on the final scRNA-seq results. However, there seems to be a lack of validation for the significance of surrogate marker genes. I am skeptical about their reliability without proper validation. More insights should be provided into this aspect.
Author Response
We appreciate the valuable comments of the reviewer.
Overall comments: In this manuscript, Kim et al. delve into the exploration of the heterogeneity present in neurons derived from directly reprogrammed neural stem cells, employing single-cell RNA sequencing. Their focus lies in comparing the distinctions between induced neural stem cells (iNSCs) and iNSC-derived neurons, revealing conspicuous heterogeneity within these cell types. The findings of this study align with existing conclusions emphasizing the prevalence of heterogeneity in directly reprogrammed neural stem cells. Regrettably, I am inclined to believe that this manuscript does not significantly contribute to our current knowledge, and therefore, I do not consider it a strong candidate for publication in Cells.
[Specific Comments]
Comment 1: I lack expertise in neuron biology, and I did not find a clear definition of "nAP-iNSC" and "AP-iNSC" in the manuscript (unless I overlooked it). Could you clarify if these terms refer to the presence of action potentials (AP) firing or spontaneous AP firing? Additionally, the reasons behind the emergence of two distinct cell groups following reprogramming are not adequately explained. Is there any clarification available?
Response 1: As you have commented above, the description might be confusing. An action potential is a main characteristic of the neurons and iNSCs that we reprogrammed did not show an action potential. However, we have named (AP)-iNSC and (nAP)-iNSC retrospectively on page 6, lines 255–261.
This means that we did not expect different expressions of directly reprogrammed iNSCs to be differentiated into two different ways, i.e., having action potentials and having no action potentials. Thus, we tried to figure out the difference and come up with the heterogeneity of neural stem cells. Thereby, we have retrospectively named the iNSCs, which express the action potential when differentiated as (AP)-iNSCs, and the others as (nAP)-iNSCs.
Comment 2: Observing Figure 1C, it appears that the expression of Ki-67 and nestin is substantially higher in (AP)-iNSC compared to (nAP)-iNSC. The localization of these markers also seems different, despite their presence in both cell groups.
Response 2: As you mentioned, it seems that the expression of Ki-67 shows differences between (AP)-iNSC and (nAP)-iNSC as evidenced in Figure 1C. However, when we counted the number of Ki-67-positive cells, the proportion of (AP)-iNSC and (nAP)-iNSC showed no significant difference.
We appreciate the reviewer’s careful and accurate comments on our ICC images. We agree with the reviewer’s comments regarding the localization of nestin expression. We found that the localization of nestin expression was different between the two groups. (nAP)-iNSC showed a radial extending morphological feature and (AP)-iNSC showed short or non-observing processes. We explained it in Results (on page 6, lines 271-273) and Discussion (on page 15, lines 454-459).
Comment 3: Is there any information regarding the relationship between AP-iNSC and AP-Neurons? According to the description in Lines 325-328, it is stated that AP-iNSC generates AP-Neurons, while nAP-iNSC generates nAP-Neurons. Could you elucidate the reasons for this distinction?
Response 3: As you have commented above, the description might be confusing. An action potential is a main characteristic of the neurons and iNSCs that we reprogrammed did not show an action potential. However, we have named (AP)-iNSC and (nAP)-iNSC retrospectively on page 6, lines 255–261.
Comment 4: If I understand correctly, the significance of marker genes lies in predicting functional neurons, and the screening of these genes is based on the final scRNA-seq results. However, there seems to be a lack of validation for the significance of surrogate marker genes. I am skeptical about their reliability without proper validation. More insights should be provided into this aspect.
Response 4: We agree that validating surrogate marker genes is necessary for future experiments.
Surrogate markers can be validated by designing short hairpin RNA or utilizing gene knockout/knockdown methods. The modified iNSCs can then be differentiated and their action potential firing can be confirmed through whole-cell patch-clamp recordings. Alternatively, we could overexpress the genes enriched in the nAP and examine if these modified cells are capable of firing. Sensitive methods, such as droplet digital PCR, can be utilized to examine gene expression levels in (nAP)-iNSCs and (AP)-iNSCs.
These validation experiments will be conducted in future studies. However, please note that our article covers more than just the simple heterogeneity of neural stem cells. Our study focused on the direct reprogramming of neural stem cells from the human fibroblasts, followed by their differentiation into neurons to explore functional heterogeneity by using single-cell RNA seq.

Reviewer 3 Report
Comments and Suggestions for Authors
The main aim of the paper is to explore the differences between of directly reprogrammed neural stem cells and the properties of the iNSC-derived neurons which have many common characteristics, but were shown by authors to be different. This study opens new directions of research of non-proliferating cells with characteristics of neurons, what is an important achievent for the whole field of research.
The paper is in excellent English, experiments are adequately described, all necessary controls are presented, electrophysiological investigation is made with correct approaches.
Pictures are informative and not excessive. Conclusions are based on presented data.
Demonstration of functional heterogeneity of directly reprogrammed induced neural stem cells (iNSCs) and iNSC-derived neurons is a new and important result.
Author Response
We appreciate the valuable comments of the reviewer.
Overall comments: The main aim of the paper is to explore the differences between of directly reprogrammed neural stem cells and the properties of the iNSC-derived neurons which have many common characteristics, but were shown by authors to be different. This study opens new directions of research of non-proliferating cells with characteristics of neurons, which is an important achievement for the whole field of research.
The paper is in excellent English, experiments are adequately described, all necessary controls are presented, electrophysiological investigation is made with correct approaches.
Pictures are informative and not excessive. Conclusions are based on presented data.
Demonstration of functional heterogeneity of directly reprogrammed induced neural stem cells (iNSCs) and iNSC-derived neurons is a new and important result.
Response: Thank you for your positive feedback, which we have taken into consideration. We appreciate your thoughtful and insightful comments and have made the minor necessary improvements to enhance the quality of the manuscript. The changes in the revised manuscript have been highlighted in yellow for your convenience.

Round 2
Reviewer 2 Report
Comments and Suggestions for Authors
I appreciate the authors for taking into consideration my suggestions and addressing my comments in a detailed and thorough manner. The revised manuscript has shown significant improvement compared to the original version.
As a final suggestion, including the appended graphical figure directly in the manuscript would enhance its clarity and completeness. This addition can provide readers with a more comprehensive understanding of the research findings and contribute to the overall presentation of the manuscript.
